# Organokines, Sarcopenia, and Metabolic Repercussions: The Vicious Cycle and the Interplay with Exercise

**DOI:** 10.3390/ijms232113452

**Published:** 2022-11-03

**Authors:** Giulia Minniti, Letícia Maria Pescinini-Salzedas, Guilherme Almeida dos Santos Minniti, Lucas Fornari Laurindo, Sandra Maria Barbalho, Renata Vargas Sinatora, Lance Alan Sloan, Rafael Santos de Argollo Haber, Adriano Cressoni Araújo, Karina Quesada, Jesselina F. dos Santos Haber, Marcelo Dib Bechara, Katia Portero Sloan

**Affiliations:** 1Department of Biochemistry and Pharmacology, School of Medicine, University of Marília (UNIMAR), Avenida Hygino Muzzy Filho, 1001, Marilia 17525-902, SP, Brazil; 2Department of Internal Medicine, Santa Casa de Misericórdia de Araçatuba, Rua Floriano Peixoto, Araçatuba 896, SP, Brazil; 3School of Medicine, Redentor University Center (UniRedentor), Avenida Presidente Dutra, Cidade Nova, Itaperuna 1155, RJ, Brazil; 4Postgraduate Program in Structural and Functional Interactions in Rehabilitation, University of Marília (UNIMAR), Marilia 17525-902, SP, Brazil; 5School of Food and Technology of Marilia (FATEC), Marilia 17506-000, SP, Brazil; 6Texas Institute for Kidney and Endocrine Disorders, Lufkin, TX 75904, USA; 7Department of Internal Medicine, University of Texas Medical Branch-Galveston, Galveston, TX 75904, USA

**Keywords:** sarcopenia, adipokines, myokines, hepatokines, osteokines, obesity, sarcopenic obesity, dyslipidemia, type 2 diabetes mellitus, exercise, metabolism, metabolic repercussions

## Abstract

Sarcopenia is a disease that becomes more prevalent as the population ages, since it is directly linked to the process of senility, which courses with muscle atrophy and loss of muscle strength. Over time, sarcopenia is linked to obesity, being known as sarcopenic obesity, and leads to other metabolic changes. At the molecular level, organokines act on different tissues and can improve or harm sarcopenia. It all depends on their production process, which is associated with factors such as physical exercise, the aging process, and metabolic diseases. Because of the seriousness of these repercussions, the aim of this literature review is to conduct a review on the relationship between organokines, sarcopenia, diabetes, and other metabolic repercussions, as well the role of physical exercise. To build this review, PubMed-Medline, Embase, and COCHRANE databases were searched, and only studies written in English were included. It was observed that myokines, adipokines, hepatokines, and osteokines had direct impacts on the pathophysiology of sarcopenia and its metabolic repercussions. Therefore, knowing how organokines act is very important to know their impacts on age, disease prevention, and how they can be related to the prevention of muscle loss.

## 1. Introduction

The estimated number of adults with diabetes mellitus (DM) in 2017 was 451 million globally, and this number is expected to augment significantly in 2035, which poses a major challenge in diagnosing, preventing, and treating this disease. Its incidence and prevalence increase as the population ages and is associated with musculoskeletal disorders and a sedentary lifestyle. Type 2 diabetes mellitus (T2DM) represents a scenario of resistance to insulin hormone, augmented advanced glycation end products (AGEs), a pro-inflammatory phenotype, and the presence of oxidative stress (OS), leading to micro- and macrovascular complications. The occurrence of diabetes is directly associated with other prevalent conditions such as obesity, dyslipidemia, and cardiovascular diseases (CVD) [1,2,3].

The World Health Organization (WHO) has designated obesity as a generalized epidemic and a chronic and progressive disease with the potential to relapse. Currently, about 2.2 billion people are overweight, representing about a third of the worldwide population (about 712 million people, representing ten percent of the global population). Instead, sarcopenia increases the fat-to-muscle ratio and is known as obesity-related sarcopenia. In National Health and Nutrition Examination Survey (1999–2004) roughly 25% of people ≥ 60 years presented sarcopenic obesity. Moreover, obesity also increases the risk of DM, dyslipidemia, and CVD [4,5,6].

Dyslipidemia is characterized by raised plasma triglycerides, total cholesterol, LDL cholesterol, or a reduced level of HDL cholesterol (HDL-c) or a combination of these features. According to a WHO estimate, the global prevalence of raised plasma total cholesterol levels among people over 25 years was 39% in 2008. That can be explained by rapid economic growth, which came with modifications in dietary habits and the adoption of unhealthy lifestyles, which have primarily contributed to the elevation of lipid levels in plasma and the prevalence of dyslipidemia and its consequences [5,7].

The metabolism imbalance reported above, as well as a sedentary lifestyle, can interfere with cell metabolism and causes cell death, leading to loss of skeletal muscle mass and strength. Suppression of insulin signaling can disrupt the phosphatidylinositol-3-kinase pathway, which promotes the reduced synthesis of proteins, impairing muscle function and integrity and contributing to the development of sarcopenia. At the same time, low muscle mass and dysfunction in sarcopenic individuals can decrease glucose clearance and reduce physical activity and metabolic rate, leading older adults with sarcopenia at increased risk for developing T2DM [1,2,3,8,9].

Physical exercise, in addition to improving lipid parameters by reducing cardiovascular risk, as well as mediating oxidative stress, might have a direct and significant impact on the prevention of sarcopenia since there is a release of myokines that promote muscle mass index preservation through a cross-link with the other organokines. However, physical inactivity leads to a lack of production of these myokines and causes the opposite effect, such as atrophy and other metabolic functions that will be explored throughout this study [10].

Irwin Rosenberg defined sarcopenia in 1988 as an age-related reduction in skeletal muscle mass and function performance. The important European Working Group on Sarcopenia in Older People, in 2010, created a practical clinical consensus for the diagnosis and proposed the reduction of muscle mass and function, including strength or performance. An update performed in 2018 defined sarcopenia as the presence of three parameters: muscle strength (assessed by handgrip strength), muscle quantity and quality (evaluated by skeletal muscle mass or appendicular lean mass), and physical performance (assessed by short physical performance battery or gait speed) as an indicator of harshness. Likewise, considering the characteristics of the local population, the Asian Working Group on Sarcopenia has developed its consensus. Decreases in muscle mass and physical performance were incorporated in the definition of sarcopenia by the International Working Group on Sarcopenia and added to the criteria created by the American Foundation for the National Institutes of Health [11,12,13,14,15,16].

The prevalence of this condition varies from 10% to 40% in healthy men and women 60 years of age or over. This is even higher in T2DM due to its association with different mechanisms such as insulin resistance, AGEs accumulation, increased OS, and vascular complications. A meta-analysis that included 6526 participants (1832 with T2DM, 4694 controls, and 1159 sarcopenia cases) reported that the sarcopenia prevalence is significantly higher in patients with T2DM than in healthy subjects. Another study showed that the prevalence of T2DM was 28.4% and 18.7% in the control group (*p* = 0.002). Similarly, older T2DM Korean men over 65 years (glycated hemoglobin (HbA1c) higher than 8.5%) were associated with low muscle quality in the lower limbs and poor physical performance. Thus, the development of a worldwide consensus that gives the definitions of sarcopenia associated with T2DM could bring benefits in the understanding, treatment, and prevention of this disease, promoting a better life quality in these individuals. During the aging process, it is possible to observe about a 30–50% reduction in the number and 10–40% decrease in skeletal muscle fibers’ size, accompanying a decrease in muscle performance [17,18,19,20,21].

In sarcopenia, there is a reduction in the number of myofibers and hypotrophic myofibers (especially type II), infiltration with adipose tissue, and after, fibrotic tissue and a reduced number of satellite cells (SCs) are seen. The etiology of the onset of this condition is multifactorial, including neurological substrates related to the loss of motor neurons, endocrine alterations resulting from the reduction of hormone expressions, such as testosterone or growth hormone (GH), reduction of muscle units, and lifestyle modifications closely linked to unhealthy nutrition and sedentary behavior [22,23,24].

Some authors could show that organokines are directly linked to homeostasis and when altered may result in profound metabolism impairment. Endocrine organs mainly produce these biomarkers, such as in muscle, adipose, liver, and even skeletal tissue, so they are directly linked with endocrine disorders such as insulin resistance, lipid disorders, metabolic syndrome, and also with sarcopenic obesity or even sarcopenia itself. The imbalance in the release of organokines can promote pro-inflammatory environments and OS, contributing to the development of disorders such as those mentioned above [25]. Due to this association and the high mortality related to metabolic imbalance, the aim of this review study was to conduct a review on the relationship between organokines (myokines, adipokines, osteokines, and hepatokines), sarcopenia, T2DM, and other metabolic repercussions.

## 2. Literature Search

The focal question for the literature survey was “How can organokines (hepatokines, adipokines, myokines, and osteokines) be on the processes related to sarcopenia manifestations and the relationship with T2DM, dyslipidemia, and obesity?”

This descriptive review used the PubMed-Medline, Embase, and COCHRANE databases. The mesh terms used were “adipokines”, “myokines”, “osteokines”, and “hepatokines” in combination with “sarcopenia”, “type 2 diabetes mellitus”, “dyslipidemia”, “obesity”, “dysmetabolic sarcopenia”, “exercise”, and “physical practicing”. The mesh terms allowed the identification of studies associated with the aims of this critical review. We did not restrict a period of time to perform the review.

## 3. General Aspects of Sarcopenia, Metabolic Repercussions, and Organokines

### 3.1. Muscular Tissue, Sarcopenia, and Physical Exercises

Muscle contraction is one of the functions of the muscular tissue, which is given by the principle of conducting chemical synapses through electrical impulses from the motor neuron to the muscle fibers innervated in the neuromuscular junction (NMJ) region. The NMJ has some central regions: presynaptic (including Schwann cells surrounding the nerve terminal with a neurotransmitter); the synaptic space lined by the basement membrane; and the postsynaptic area (with the junctional sarcoplasm) and the postsynaptic membrane containing receptors for the neurotransmitter [26,27].

As pointed out above, during the aging process, there is a reduction in the number and a decrease in the size of skeletal muscle fibers. Mitchell et al. [28] found a reduction in skeletal muscle mass of about 0.37 and 0.47%/year in women and men, respectively. Other clinical studies involving aging humans indicated that muscle strength decreases in a higher way in older adults than muscle mass. With aging, changes in structural proteins may occur, causing reduced muscle performance and alteration of proteins, such as skeletal muscle cells’ calcium handling proteins. In addition, aging can also promote a decline in muscle performance due to the imbalance of excitation–contraction duet, an increase in intermuscular fat tissue, and the quantity of extracellular water associated with the muscle volume [29,30].

Sarcopenia is the condition in which the muscles present a reduced number of myofibers and hypotrophic myofibers (mainly type II myofibers), infiltration with adipose tissue, and, in later stages, fibrotic and a decrease in the number of satellite cells (SCs). The etiology of the onset of sarcopenia is multifactorial, involving neurological factors related to the loss of motor neurons, endocrine changes due to reduction or loss of hormone expression, such as testosterone or growth hormone (GH), loss of muscle motor units, and also by nutritional and lifestyle changes closely linked to adherence to sedentary habits [22,23,24].

The pathogenesis of sarcopenia can lead to several molecular dysregulations. Forkhead box O3 (FoxO3)-dependent transcription and reduced protein production positively regulated by insulin-like growth factor-1 (IGF-1) and via phosphoinositide 3-kinase/protein kinase B (PI3K/AKT) lead to reduced muscle size due to smaller cell size (loss of cell content) resulting from catabolism of the protein through amplified proteasomal and lysosomal functions. This process also reduces energy production, leading to fatigue. FoxO3a leads to iron accumulation, as it leads to increased expression of human ferritin and positively regulates iron transport, increasing motor neuron vulnerability to degeneration and negatively affecting myokine production. Thus, there may be a two-way pathway, as atrophy and reduced physical performance may be due to myokine signaling that is modified during sarcopenia, and at the same time, the reduction and expression of myokine generation are due to muscle failure [31,32,33].

Myokines and interleukin IL-15 are capable of stimulating myofiber hypertrophy, IL-6 increases glucose uptake and fatty acid oxidation, IL-8 can lead to the stimulation of angiogenesis, and they have a relationship with age-related alterations in muscle function and strength. TNF-α activates endothelial cells and the synthesis of nitric oxide (NO), increasing vascular permeability, promoting the release of pro-inflammatory cells, and triggering the inflammatory process. As a result, it is an important marker of muscle loss and muscle strength during the latent inflammatory process. The activation of apoptosis in muscle cells can also be stimulated by this organokine. The association of IL-6 and TNF-α results in the liver production of acute-phase reactive proteins (C-reactive protein (CRP) or α1-antichymotrypsin (ACT)). The increased concentration of IL-6 and CRP results in a decrease of physical performance and disability. On the other hand, macrophages modulate the regenerative muscle response associated with a decrease/inhibition of the regenerative muscle capacity, also due to the greater degree of neutrophilia and the creation of a pro-inflammatory environment [34,35,36].

Furthermore, sarcopenia involves an upregulation of myogenin, an essential transcription factor in regulating myoblast differentiation. Myogenin has been found to induce muscle wasting under different conditions, including denervation, spinal muscle atrophy, starvation, and tumor necrosis factor α-induced atrophy (TNF-α). In conjunction with increased myogenin, higher levels of Heat Shock 70kDa protein 1 (HSPA1A) and lower production of nuclear factures erythroid 2-related factor 2 (Nrf2) may contribute to intracellular accumulation of ROS, negatively impacting tropism and cell function. ROS imbalance may determine the unusual activity of p38 mitogen-activated protein kinase (MAPK) and dysregulated expression of the cell proliferation inhibitor, p16INK4a, disrupting the Janus kinase/signal transducer and activator of transcription (JAK-STAT) signaling and defective autophagy detected in aged SCs, responsible for their altered proliferation and differentiation capacities [37,38,39,40,41].

Finally, some neuronal factors are also involved. Brain-derived neurotrophic factor (BDNF) is an example. It is a neurotrophin associated with synaptic plasticity and cell survival observed with the aging process. BDNF is stimulated by the contraction of skeletal muscles resulting in augmented fat oxidation and muscle fiber actions that occur in the contraction events. However, the lack of BDNF can lead to muscle atrophy due to the reduced capacity to produce proteins. It is also associated with the reduction of myogenic regulatory factors. In this context, it was observed that the synthesis and release of BDNF in the brain can increase with physical activity, delaying the onset of sarcopenia [42,43,44,45].

Figure 1 represents the main pathophysiological pathways observed in the occurrence and progression of sarcopenia.

### 3.2. Sarcopenia and Diabetes

In sarcopenia, a phenomenon occurs in which the loss of skeletal mass and changes in fat deposition can promote metabolic complications since the accumulation of visceral and intramuscular fat can be observed at an increased risk of cardiovascular disease, insulin resistance, and diabetes [46]. Insulin resistance, especially in skeletal muscle and the liver, and defective insulin secretion by the pancreas are the two main pathophysiological mechanisms observed in T2DM. However, there is a set of diverse factors that involve T2DM resulting from an interaction between genes and the environment, as there is growing evidence that the risk of T2DM is strongly influenced by genetic factors [47,48,49].

Insulin resistance in diabetes is associated with chronic inflammation through the release of pro-inflammatory cytokines and microparticles and can trigger the activation of innate immune cells, inducing a vicious cycle of inflammation in various tissues and insulin resistance [50,51].

The pro-inflammatory scenario observed in T2DM patients results in systemic inflammation directly related to sarcopenia, as they share chemical mediators related to the pathophysiology of the two conditions. Park et al. [52] demonstrated that older adults with T2DM had a higher loss of muscle mass and leg strength over three years than non-diabetic controls. These associations were partially attenuated after adjustment for cytokines, including IL-6 and TNF-α. Moreover, Visser et al. [53] assessed 2746 older adults in good general condition aged between 70 and 79 years, and they noted that there was a decrease in handgrip strength between 1.1 and 2.4 kg in the presence of an increase in the concentration of IL-6, revealing a decline in muscle strength in these conditions. In the English Longitudinal Ageing Study, increased CRP was negatively associated with handgrip strength only in women and lower body strength in both sexes. Therefore, it is concluded that the inflammation, in vivo, associated with T2DM can affect muscle mass and strength [54].

Chronic complications of T2DM, such as neuropathy, retinopathy, nephropathy, other microvascular complications, and innervations that end up reaching the muscles, will result in the reduction of muscle contractility and, consequently, loss of muscle strength. OS, together with the inflammatory environment, promotes endothelial dysfunction, which can lead to peripheral arterial disease (PAD). This extra macrovascular complication affects up to a quarter of patients with T2DM. The blood pressure reduction can lead to ischemia, decreasing the results of strength, mass, and muscle performance. Furthermore, due to the pain associated with this disorder, PAD can lead to a decrease of physical activity and exercise, further contributing to the reduction of muscle health [25,54,55,56].

In summary, the available pieces of evidence reviewed show that sarcopenia occurrence in patients with T2DM is increased. As pointed out previously, numerous direct and indirect links between T2DM and sarcopenia involve insulin resistance, inflammatory factors, AGE accumulation, oxidative stress, and macro- and microvascular complications that can affect metabolism, regeneration, and muscle strength in several ways. Moreover, the existence of one condition can increase the risk of developing the other [56].

### 3.3. Sarcopenia and Obesity

When approaching aging and obesity, it is understood that there is an increase in life expectancy along with a lifestyle change. The aging process is associated with decreased muscle mass and strength and increased body fat mass, leading to disability, frailty, falls, social isolation, and possible hospitalization. Therefore, the designation of sarcopenic obesity (SO) appears to also represent the presence of sarcopenia and obesity [57].

Several factors cause OS. We can cite the age events, reduced physical activity, unhealthy nutrition, low-grade inflammatory processes, resistance to insulin, and hormonal changes which lead to changes in body composition. Aging reduces basal metabolic rates leading to weight gain and a decrease in muscle mass, as shown by in vivo studies [58,59]. Older people typically reduce physical activity, contributing to muscle strength loss and resulting in atrophic muscles that further aggravate the lack of physical performance. Furthermore, in the elderly, there is a significant imbalance between energy intake and expenditure, which is also related to OS and an increase in inflammatory processes in in vivo studies. Adipocytes promote macrophage recruitment, so there is an augmented organokine secretion by adipocytes and immune cells (leptin, resistin, and chemerin) and other pro-inflammatory markers (TNF-α, interleukins, and interferon-γ), creating a circumstance of low-grade inflammation which plays a significant role in OS progression. In addition to being related to low-grade chronic inflammation, insulin resistance is also correlated with mitochondrial dysfunction, decreasing muscle strength and increasing fat accumulation in muscle and the liver. Lastly, the disruption in the release of organokines and reduced concentrations of testosterone and estrogen are important factors related to the aging process [60,61,62].

In addition, in sarcopenia, there is an increase in IL-6, which is produced by macrophages and stimulated by the stimulation of the nuclear factor κ-light-chain-enhancer of activated B cells (NF-κB). NF-κB is capable of reducing IRS1 (insulin receptor substrate 1) and GLUT-4 (glucose transporter type 4) expression in tissues such as adipocytes, promoting a pro-inflammatory environment. Furthermore, a reduction of adiponectin in plasma levels contributes to the increase of insulin resistance and the inflammatory pattern of adipocytes [25,63,64].

Figure 2 shows the most important events that occur during sarcopenic obesity development.

### 3.4. Sarcopenia and Dyslipidemia

Dyslipidemia and diabetes are closely interlinked mainly due to their pathogenic mechanisms; therefore, they are also interlinked with obesity and sarcopenia. In recent decades, several mechanisms have been shown to contribute to the worsening of atherosclerosis in patients with insulin resistance, which increases endothelial cell dysfunction and thus decreases the bioavailability of nitric oxide, which is a potent vasodilator. In addition, a review suggested that hyperglycemia contributes to glucotoxicity and exerts a synergistic pro-atherogenic effect in the vascular bed alongside dyslipidemia and hypertension [65].

As mentioned previously, with aging, there is a decrease in hormone production. In menopause, for example, there is an increase in cardiovascular risk due to estrogen deficiency and unregulated lipid metabolism. Estrogens play a protective role in the cardiovascular system and can be produced in the ovaries using LDL-c. However, circulatory LDL-c is not used to synthesize estrogen during menopause, leading to a decrease of estrogen production. Therefore, some reviews suggested that menopause is associated with high levels of LDL-c and reduced estrogen and testosterone levels, leading to a reduction in muscle mass and muscle strength [60,66].

Insulin resistance promotes the increase of glycogenesis, increase of the expression of sterol regulatory element-binding protein 1c (SREBP-1c), inhibition of β-oxidation, increase of free fatty acids supply, and alters the transport of triglycerides, resulting in the accumulation of triglycerides in skeletal muscle and the liver [29,30]. Habib et al. [30] analyzed a population of 288 adult male individuals to conclude that total cholesterol and triglycerides levels were significantly higher, and HDL-c was significantly lower in sarcopenic obese subjects compared to non-sarcopenic obese subjects, showing that sarcopenia aggravates dyslipidemia.

Finally, a Korean study reported that regardless of abdominal obesity, people with a lower skeletal muscle mass index were significantly associated with a higher risk of dyslipidemia. The authors conclude that preventing muscle wasting may be an interesting strategy to manage LDL-c levels, which further proves the need to intervene in sarcopenia as a whole, since it is directly associated with classic features of metabolic dysfunction that may reflect pathologies at vascular levels in the long term [66,67].

### 3.5. Organokines and the Relations with Sarcopenia, DM, Sarcopenic Obesity, and Dyslipidemia

Sarcopenia and its metabolic implications may have a pathophysiological mechanism at the molecular level, which brings us to organokines, as they are involved in mechanisms such as insulin resistance, T2DM, obesity, metabolic syndrome, and CVD. These molecules are being increasingly investigated and can be found in skeletal, adipose, and liver cells that, respectively, release myokines, adipokines, and hepatokines. Thus, it is noted that organokines can act as showing benefits or harmful effects to the organism and even perform the known crosstalk when it acts in endocrine, paracrine, and autocrine pathways. Table 1 shows the most studied organokines involved in sarcopenia and its metabolic repercussions. In turn, Figure 3 shows the possible mechanistic associations between organokines secretion and the occurrence of dysmetabolic sarcopenia.

#### 3.5.1. Myokines

##### Irisin

Irisin is a designated myokine produced from the proteolytic cleavage of fibronectin type III domain, and its secretion follows physical exercise. It stimulates osteogenesis by integrin actions and promotes bone mass regulation by inducing osteoblasts’ emergence, decreasing osteoclasts activation and increasing cortical bone mass in mice. In high concentrations, it trans-differentiates white adipose tissue into brown adipose tissue, possibly promoting weight loss, improving metabolic syndrome parameters, improving insulin sensitivity, and protecting against the development of cardiovascular diseases. Irisin is usually secreted during and after physical exercise and strength training [33,108,110,111].

Independently, irisin is a myokine that correlates with sarcopenic obesity in decompensated T2DM patients. At low levels, this molecule also leads to increased insulin resistance [68]. Among Chinese individuals, Liu et al. [69] found in a cross-sectional study that serum irisin levels are inversely correlated with dyslipidemia. Lack of exercise is the main cause of lower irisin levels, which connects intimately with protein and muscle mass loss in the occurrence of sarcopenia. Sarcopenic individuals with a lack of physical activity may have the effects of irisin aggravated since irisin can develop insulin resistance by enhancing type 4 glucose transporter (GLUT4) translocation and b-oxidation of free fatty acids via energy sensor AMP-activated protein kinase [113].

##### Myostatin

Myostatin is a molecule that is part of the transforming growth factor (TGF-β) superfamily expressed mainly by skeletal muscle cells of humans. In its physiological effects, this myokine under-stimulates muscle construction while limiting muscle growth and promoting protein breakdown. Due to these effects, myostatin leads to muscle atrophy and muscle cachexia principally by acting through the modulation of extracellular binding proteins of transcriptional and epigenetic regulation. Recently, myostatin was also implicated in developing obesity, insulin resistance, and cardiovascular diseases [114,115].

Among sarcopenic individuals, this myokine contributes to the development of sarcopenic obesity. In recent studies, myostatin was correlated as a predictor of sarcopenic obesity when at higher levels. Being inversely proportional expressed to skeletal muscle mass in sarcopenia, myostatin promotes augmented muscle growth impairment and protein degradation more than in other individuals. These effects contribute to impaired glucose metabolism due to increased insulin peripheral resistance. Furthermore, there is a hypothesis that serum myostatin level can be used as a monitoring biomarker for frailty and sarcopenia diagnosis [71,72].

Figure 4 shows the intracellular signaling pathways involved in the myostatin regulation of protein degradation during sarcopenia.

##### Sclerostin

Sclerostin is a circulating protein from osteocytes that is related to the formation and regulation of bone mass, and its deficiency leads to hereditary conditions of high bone mass characterized by excessive bone formation. Studies suggest that non-circulating sclerotin acts as a biomarker in non-skeletal diseases [73,74].

Another effect occurs when sclerostin is elevated in patients with metabolic disorders. Insulin, which has an osteogenic impact, directly increases the number and function of osteoblasts and indirectly controls blood glucose levels. As a consequence, there is a curtailment in bone mass in individuals with diabetes due to a defect in insulin production/resistance. Hyperglycemia leads to a decrease in bone formation, as it directly increases sclerostin production, inhibiting bone formation by downregulating the Wnt pathway (a signaling pathway that affects bone modeling and remodeling). In addition, there is a relationship between elevated sclerostin levels with the increase in the risk of spinal fracture [75,112,116,117].

Only a few studies merge sclerostin with sarcopenia. Ahn et al. [76] demonstrated in a population of 129 senior participants that this protein, when at higher levels, is significantly associated with a lower risk of sarcopenia, even considering the confounding effects of genera, age, and body weight. Therefore, sclerostin may be a potential new biomarker for sarcopenia [73,76].

##### Brain-Derived Neurotrophic Factor (BDNF)

Brain-derived neurotrophic factor (BDNF) is considered a neurotrophin directly associated with memory and learning found in the brain and skeletal muscles. BDNF, under normal conditions, acts mainly on muscle injury, which works on satellite cells and modulates their response in muscle regeneration. As expressed in muscle satellite cells, when found in low concentrations, it negatively affects the myogenesis process. In addition, BDNF expression increases with physical exercise. Obese and T2DM patients present BDNF-reduced levels [77]. There is also an association in relation to the overexpression of BDNF due to the augment in the number of glycolytic fibers. When produced by muscle, it increases insulin sensitivity and also acts in glucose and lipids regulation [25,118,119].

A Japanese study on hemodialysis patients revealed that low levels of BDNF were associated with a decrease in muscle function and an increased prevalence of severe sarcopenia [78]. At the metabolic level, the BDNF index in skeletal muscle was decreased in mice bearing T2DM [120]. Due to the above data, it has been suggested in reviews that BDNF expression plays an important role in muscle aging, which may have implications on the pathogenesis of sarcopenia and its metabolic repercussions [79,80].

##### Interleukin-6

Interleukin-6 (IL-6) is a dual organokine, as it is secreted by both myocyte and adipocyte. IL-6 is directly related to BMI by acting on adipocytes and is increased in individuals with insulin resistance, obesity, and T2DM [10,25,81,82]. IL-6, as a myokine, acts by regulating satellite cells leading to skeletal muscle hypertrophy. It also upregulates anti-inflammatory biomarkers and inhibits the pathways related to the TNF-α activity.

As a pro-inflammatory factor in obese people, it leads to a decrease in IGF-1 levels and a reduction in muscle volume and strength. Some other factors which act together with IL-6 can also promptly inhibit phosphoinositide 3-kinases (PI3Ks)/protein kinase B (Akt) actions and protein synthesis pathways such as TNF-α, IL-1B, and IL-1. In this way, skeletal muscle is reduced through its receptor signaling [10,79,81,91]. For confirmation, Scheffer and Latini [121] showed in a review that decreased strength in resistance training is associated with increased plasma levels of pro-inflammatory factors (TNF-α, IL-6, and IL-8). In diseases such as sarcopenic obesity and others involving conditions of persistent inflammation, IL-6 is linked to muscle atrophy [10].

On the other hand, IL-6 can augment the oxidation of triglycerides in muscle cells and plasma glucose and induces lipolysis in adipose tissue during exercise [33]. In general, when performing physical exercises, IL-6 (as a myokine) can increase hypertrophy and leads to lipolysis. Moreover, it provides an anti-inflammatory scenario. Moreover, a decrease in plasma IL-6 results in chronic inflammation in patients with inflammatory and metabolic conditions [10,79,81].

##### Interleukin-15

The actions of IL-15 occur under the mediation of Janus kinase (JAK)/signal transducer and activator of the transcriptional protein signaling pathway (STAT), as well as PI3K/Akt and AMP-activated protein kinase (AMPK) signaling pathway. It belongs to the IL-2 superfamily and acts in muscle–adipose tissue crosstalk. This organokine is released through physical exercise in a pulsatile and short duration and can reduce the mass of visceral adipose tissue in humans and rodents. Yang et al. (Yang et al., 2013) found low levels of IL-15 in the plasma and muscle of animals that were fed a high-fat diet. IL-15 is an anti-inflammatory factor, mainly due to the downregulation of the TNF-α expression, which acts on oxidative stress. In addition to preventing the reduction of muscle mass, it increases glucose uptake by the skeletal muscle by stimulating the mobilization of GLUT 4, which leads to a contribution to muscle hypertrophy and reduces subcutaneous fat [10,80,83,84,85].

Regarding the pathogenesis of sarcopenia, Sakuma and Yamaguchi [72] have suggested in their review that IL-15 levels are low in individuals with sarcopenia and that its levels gradually decline with aging. In humans, it is possible to verify the decline of AMPK actions with aging. Furthermore, IL-15 can counteract obesity induced by high-fat diets, insulin resistance, and fat liver by inhibiting lipid accumulation in preadipocytes and adiponectin secretion, indirectly reducing adipose tissue mass [80,84,93].

#### 3.5.2. Adipokines

##### Leptin

Leptin is one of the most secreted adipokines derived from adipose cells. This molecule belongs to the class-I helical cytokine superfamily, and its effects derive from the activation of intracellular signal transducers and transcription signaling pathways. One of the major functions of leptin is the regulation of energy homeostasis and the amount of food intake, despite having roles in autoimmunity, being associated with both adaptative and innate immune systems. Leptin metabolic actions mainly increase lipolysis and the skeletal muscles’ insulin sensitivity [122,123]. In low-grade inflammatory processes, leptin is capable of inducing the synthesis of both IL-6 and TNF-α inflammatory cytokines and has its secretion augmented in the presence of both IL-1 and TNF-α cytokines. In liver diseases, leptin correlates with non-alcoholic fatty liver diseases [111]. Among sarcopenic individuals, leptin serum levels are elevated. However, when this adipokine is reduced in sarcopenia, it has been implicated in the occurrence of sarcopenic obesity. Additionally, in previous studies, lower levels of leptin in sarcopenia were correlated with the induction of insulin resistance among peripheral tissues, principally in skeletal muscles, contributing to diabetes occurrence among sarcopenic individuals. These actions decrease glucose consumption in the muscles, probably decreasing their mass, which also can contribute to the development of sarcopenic obesity [86,87].

##### Lipocalin 2

Lipocalin 2 (LCN2) is a glycoprotein present as a cytokine in the adipose tissue of obese individuals with metabolic syndrome when there is low-level systemic inflammation. It is associated with neutrophil gelatinase-associated lipocalin (NGAL), and its receptors regulate inflammation and transport fatty acids. LCN2 functions are part of the innate immune system and can reduce the growth of pathogenic bacteria via iron sequestration. LCN2 is also involved in the maintenance of iron homeostasis [25,88,89].

It has been shown that iron accumulation mediated by this cytokine is upregulated in obese mice with sarcopenia. It is therefore suggested that the synthesis of these iron regulatory proteins in sarcopenic obesity contributes immensely to inflammation and oxidative stress [90]. In addition to promoting anti-inflammation, it protects against cellular and tissue stress. There may be an association of the tissue level of LCN-2 with insulin sensitivity and glucose homeostasis as it is increased in obesity and type 2 diabetes [89].

##### Interleukin-6

As shown in topic 4.5.1, IL-6 is a dual organokine secreted by both myocyte and adipocyte. As an adipokine, IL-6 is increased in individuals with obesity and metabolic syndrome. IL-6 is directly related to inflammation and the inhibition of the expression of insulin receptor substrate 1 (IRS1) and GLUT4 in adipocytes. Its levels are increased in individuals with pro-inflammatory conditions such as insulin resistance, obesity, sarcopenic obesity, and T2DM [10,25,81,82].

##### Interleukin-10

IL-10 is released by TH cells, monocytes, and macrophages. It can regulate neutrophil activity and plays an important role in macrophage switching from M1 to M2 in injured muscle, which is necessary for the growth and regeneration of muscle tissue. Moreover, this interleukin avoids inflammation since it can suppress macrophage activation as well as interferon-gamma (IFN-γ), TNF-α, IL-2, and IL-6. IL-10 actions are related to the reduction of inflammatory events, obesity, and oxidative stress in skeletal muscle [10,79].

Hacham et al. [124] showed an increase in IL-10 levels in older mice, and Álvarez-Rodriguez et al. [92] registered an augment in the IL-10 levels in older humans. Finally, it was reported that in individuals with sarcopenic obesity, physical exercise releases IL-10 from skeletal muscle to the bloodstream, which generates a more anti-inflammatory environment and induces protein synthesis [10,72].

Pfeiffer et al. [125] performed the pre-incubation in vitro of IL-10 in islets of human pancreatic beta cells, which did not preserve the function of beta cells but decreased the infiltration of T cells that would lead to the autoimmune mechanism against the islets. The experiment showed positive results in diseases such as autoimmune diabetes. Unexpectedly, IL-10 can also be associated with insulin resistance in obese subjects due to the suppression of adipocyte energy expenditure and thermogenesis. CD4+Foxp3+ regulatory T cells (Tregs) represent a critical source of IL-10, where the loss of IL-10 Treg resulted in increased insulin sensitivity and reduced obesity in mice fed with a high-fat diet [126].

##### Interleukin-15

IL-15 is also considered an adipokine. It presents anti-inflammatory properties and acts on oxidative stress, preventing the reduction of muscle mass and reducing subcutaneous fat. It also has a role in the inhibition of lipid accumulation in preadipocytes and adiponectin secretion, which indirectly reduces adipose tissue mass [10,80,85,93].

##### Apelin

Apelin can act as an adipokine and myokine and is widely present in many tissues. It operates through the apelin/APJ system, being an endogenous binder of APJ receptor (G protein-coupled receptor). Its release is regulated by physical exercise, which improves metabolism in obesity and T2DM [80,95,127].

In addition, there is a significant decline in apelin levels with aging, which leads to a reduction in muscle function. On the other hand, an augmented level of apelin improves muscle function as it acts in an anti-inflammatory way and increases the regenerative abilities of muscles, which indicates that apelin may be a biomarker of early sarcopenia and even a possible treatment in this disease [10,80]. To finalize, Afshounpour et al. [94] carried out a resistance and aerobic training program among 24 male patients with T2DM for eight weeks and showed a significant increase in apelin amounts and found diminished glycemia and insulin resistance.

##### Insulin-like Growth Factor Hormone (IGF-1)

IGF-1 is known as a growth factor hormone produced mainly by the liver in adults, acting as a systemic agent, but it is also synthesized by the skeletal muscle acting in an autocrine and paracrine way. IGF-1 promotes anabolism through PI3K/Akt/mTOR and PI3K/Akt/GSK3β pathways and acts as an important agent in muscle growth, differentiation, and regeneration since it improves myogenesis and increases the strength of myofibers [10,107]. This mechanism occurs through the stimulation of satellite cells, contributing to muscle hypertrophy. In disorders such as sarcopenia, IGF-1 signaling and the IGF-1 index are suppressed, resulting in muscle atrophy resulting from the combined effects of altered protein synthesis, autophagy, and impaired muscle regeneration [10,80].

Plasma IGF-1 decreases with age, which implies a risk for the development of sarcopenia since there is a hypothesis that low IGF-1 is a possible biomarker for sarcopenia [95,128]. Poggiogalle et al. [97] analyzed patients between 18 and 65 years old to show that the GH/IGF-1 axis deficiency is related to a greater chance of developing sarcopenic obesity. Gut microbiota may also have a role in the pathogenesis of sarcopenia since short-chain fatty acids (SCFA), which are the end products of anaerobic fermentation of the intestinal microbiota, are the central modulators of IGF-1 synthesis. Therefore, this fact may directly link skeletal muscle and gut microbiota [96,129].

##### Fibroblast Growth Factor 21

Fibroblast growth factor 21 (FGF-21) can regulate glucose and lipid metabolism, inhibiting lipogenesis and stimulating the expression of adiponectin in the bloodstream. It also has antioxidant and immunological functions and can induce bone resorption [25,130].

FGF-21 improves diacylglycerol levels in the liver and skeletal muscle and inhibits protein kinase C translocation, which can lead to insulin resistance. Suppression of FGF-21 can avoid the browning of the white adipose tissue. In addition, there is an association in the reduction of hepatic synthesis of FGF-21 in obesity, which fails to stimulate glucose uptake and reduces mitochondrial activity and thermogenesis [96,131,132].

In people with obesity, the concentration of FGF-21 increases with physical activity one hour after exercise, which can prevent some diseases, mainly sarcopenia and obesity [133]. When increasing energy consumption in response to exercise, the body increases energy sources via mitochondria through FGF-21 and irisin, explaining the strong correlation between FGF-21/irisin and muscle mass [10,83,98].

##### Adiponectin

Adiponectin is considered an anti-inflammatory protein initially secreted by adipose tissue but is also produced by skeletal muscle. It is made up of 244 amino acids that mainly act on the liver and skeletal muscle cells. In health and against CVD, adiponectin exerts anti-inflammatory actions in obesity, type 2 diabetes mellitus, metabolic syndrome, and atherosclerosis, increasing insulin sensitivity. At the muscle level, there is an increase in free fatty acid oxidation and glucose uptake; at the liver level, adiponectin reduces gluconeogenesis [80,99,134].

With regards to anti-inflammatory action and energy homeostasis, adiponectin enhances myogenesis, avoiding muscle protein degradation through regulation of the IRS-1/Akt signaling pathway [135]. More specifically, the reduction of inflammation occurs by strongly reducing TNF-α and IFN-γ synthesis and increasing the production of IL-10 and IL-1 antagonists of monocyte and macrophage receptors. For these reasons, adiponectin could be used to reverse sarcopenia by inhibiting anti-atrophy proteins and stimulating proteins in mice [136]. Priego et al. [135] reported in their review that the level of adiponectin in people with sarcopenia is low. However, there is an inverse response to this fact in elderly individuals through a balance in response to maintaining metabolic homeostasis and reducing inflammatory processes, oxidative stress pathways, and catabolic conditions such as sarcopenia [10,81].

Sarcopenic obesity is associated with increased adipose tissue mass due to the association with large changes in adipokines, such as leptin and adiponectin, and resistin/IGF-1 ratio [108]. It was observed in obese people that there was decreased TNF-α and IL-6 after 12 weeks of exercise, while the levels of adiponectin and IL-10 increased [137]. Furthermore, the weight loss resulted from the practice of physical exercises, which can increase plasma adiponectin levels both in rodents and humans, as adiponectin released by physical practice can provoke stimulation in the oxidation of fatty acid and glucose uptake through the AMPK pathway, which is highly compromised in sarcopenic obesity patients [138]. Thus, increased adiponectin levels can lead to an increase in muscle hypertrophy, a decrease in inflammation, and an increase in fat oxidation. In this way, physical exercise can be evaluated as a non-invasive golden intervention for sarcopenic obesity [10].

#### 3.5.3. Hepatokines

##### Fetuin-A

Fetuin-A, a natural insulin receptor tyrosine kinase inhibitor, is an endogenous binder for Toll-type 4 receptors through which saturated fatty acids induce pro-inflammatory signaling and insulin resistance. Fetuin-A has been shown to have a high association with non-alcoholic fatty liver disease. This metabolic pattern occurs by virtue of the increase in the amounts of fetuin-A interfering with the insulin signaling cascade and the translocation of GLUT-4 in the insulin target tissues and, as a consequence, causing damage to the pancreatic β cells [69,100,101].

Chang et al. [102] conducted a study involving 541 adults aged 65 and over in a Taiwanese community and hypothesized that fetuin is likely linked to heart hypertrophy. This molecule, a systemic inhibitor of calcification, is increased in sarcopenic patients. In contrast, it is decreased in sarcopenic patients with ventricular hypertrophy, demonstrating a curious opposition. Even though it is associated with sarcopenia, its mechanism still needs to be better investigated for more accurate results. This hepatokine builds a pro-inflammatory scenario from the polarization of M1 macrophages in the white adipose tissue, producing reactive oxygen species and decreasing insulin action, which, together with other pro-inflammatory cytokines, contribute to a state of obesity, sarcopenia, and sarcopenic obesity [25,96,102,139,140].

##### Sex Hormone Binding Globulin

There are few studies relating sex hormone binding globulin (SHBG) with sarcopenia and their metabolic effects. However, it is known that SHBG is a glycoprotein synthesized mainly in the liver that binds and transports sex steroids to their target tissues. At the metabolic level, its serum levels are inversely associated with the occurrence of diabetes and fatty liver diseases and anti-inflammatory actions [99,141,142].

Low SHBG concentration was analyzed in some reviews/crosstalk, and it was identified in T2DM and obese patients. On the other hand, modifications in lifestyle, such as reduction of body weight, caused an increase in circulating SHBG. Therewithal, adiponectin increases SHBG production by activating AMPK and reducing the activity of enzymes involved in hepatic lipogenesis. However, the relationship between SHBG and increased insulin sensitivity still has to be unraveled [25,143].

##### Leukocyte-Derived Chemotaxin-2

Leukocyte-derived chemotaxin-2 (LECT2) is a hepatokine that has action over hepatic β-catenin. Also known as chrondomodulin-2, it was identified only as a promoter of neutrophil migration to inflamed tissues. This hepatokine has been correlated with regulatory actions on obesity, glucose metabolism, and immunological and inflammatory processes, in addition to the development of non-alcoholic fatty liver diseases [99,106,144].

LECT2 acts directly on skeletal muscle by positively correlating insulin resistance and obesity [96,145]. Yoo et al. [105] revealed in vivo that serum levels of LECT-2 are increased in individuals with obesity and fatty liver disease, in addition to promoting N-terminal Jun kinase (JNK) phosphorylation in myocytes, leading to impaired insulin sensitivity in mice and an intercorrelation with metabolic syndrome.

To our knowledge, there are no studies that show that LECT-2 is directly linked to sarcopenia. However, since this molecule has direct implications for skeletal muscles as well as its metabolic actions, the possibility of a future discovery of this relationship should not be ruled out.

##### Insulin-like Growth Factor Hormone (IGF-1)

As shown in topic 4.5.2, IGF-1 is also a hepatokine, in which a lack of it can imply a risk for the development of sarcopenia by the mechanism of muscle hypertrophy and atrophy inhibition. Moreover, there are other metabolic implications, such as sarcopenic obesity and direct impacts on oxidative stress [10,95,107].

#### 3.5.4. Osteokines

##### Osteocalcin

The osteocalcin (OCN) is an osteokine synthesized and secreted mainly by osteoblasts, known to increase in acute episodes of physical exercise and presents itself in three forms: carboxylated, subcarboxylated and non-carboxylated. The uncarboxylated or subcarboxylated forms (ucOCN) have been associated with several metabolic effects, as they can increase insulin secretion and sensitivity and act in the direct uptake of glucose into skeletal muscle. In addition, OCN allows the survival and function of pancreatic β cells, thereby increasing the production of insulin. What is interesting is that insulin itself can induce the release of OCN [108,109,146].

OCN mediates two other metabolic mechanisms. First, this hormone also establishes positive feedback with IL-6, resulting in an increase in glucose utilization. The second occurs because, when there is a set of glucose consumption by the skeletal muscle, in addition to being mediated by IL-6, OCN decreases the substrates available for lipogenesis and can also induce the appearance of an anti-inflammatory environment. These phenomena act indirectly to reduce visceral fat mass. Finally, from a functional perspective, this biomarker also contributes to muscle hypertrophy and strength [99,108,109].

##### Irisin

As shown before, this molecule stimulates osteogenesis by integrin actions. It promotes the regulation of bone mass by inducing the emergence of osteoblasts, decreasing osteoclasts activation, and increasing cortical bone mass. In addition, it improves insulin sensitivity, protects against the development of CVD, and promotes muscle mass loss in lower levels [33,108,110,111].

##### Sclerostin

As previously mentioned, sclerostin is also an osteokine that is related to the formation and regulation of bone mass, and its deficiency leads to hereditary conditions of high bone mass. It reduces bone mass in individuals with diabetes, as there is a defect in insulin production/resistance. Its excess is related to metabolic disorders, while insulin increases the number and function of osteoblasts, which indirectly controls blood glucose levels. Therefore, it is being argued that sclerostin has great potential to directly implicate sarcopenia and other metabolic implications [73,75,76,112].

## 4. Conclusions

Sarcopenia is a condition related to the aging process and has a strong association with pro-inflammatory and oxidative environments. The risk of fractures due to decreased muscle mass and strength, cardiovascular events, and quality of life can directly impact the prognosis if preventive measures are not taken, such as, mainly, the practice of physical exercises throughout the course of life.

Organokines help to unravel the molecular universe of sarcopenia and to open paths for the understanding of its pathophysiology, which often makes a crosstalk acting in more than one tissue at the same time, thus bringing answers about how the increase or reduction of some myokines, adipokines, hepatokines, or osteokines can influence the improvement of the clinical scenario of patients affected by sarcopenia.

Given the above, the understanding of organokines can help in the rehabilitation and prevention of sarcopenia and its consequences, as well as reiterating the importance of physical exercise due to its optimal impact on the disease.

## Figures and Tables

**Figure 1 ijms-23-13452-f001:**
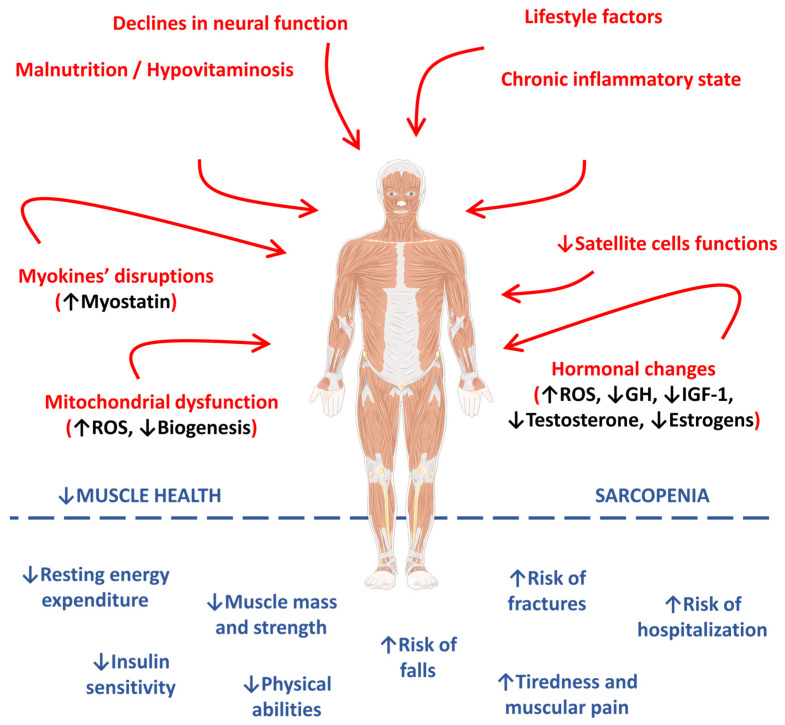
Main pathophysiological pathways involved in the occurrence and progression of sarcopenia. ↑, increase; ↓, decrease; GH, growth hormone; IGF-1, insulin-like growth factor 1; IL-1β, interleukin 1 beta; IL-6, interleukin 6; ROS, reactive oxygen species; TNF-α, tumor factor necrosis alpha.

**Figure 2 ijms-23-13452-f002:**
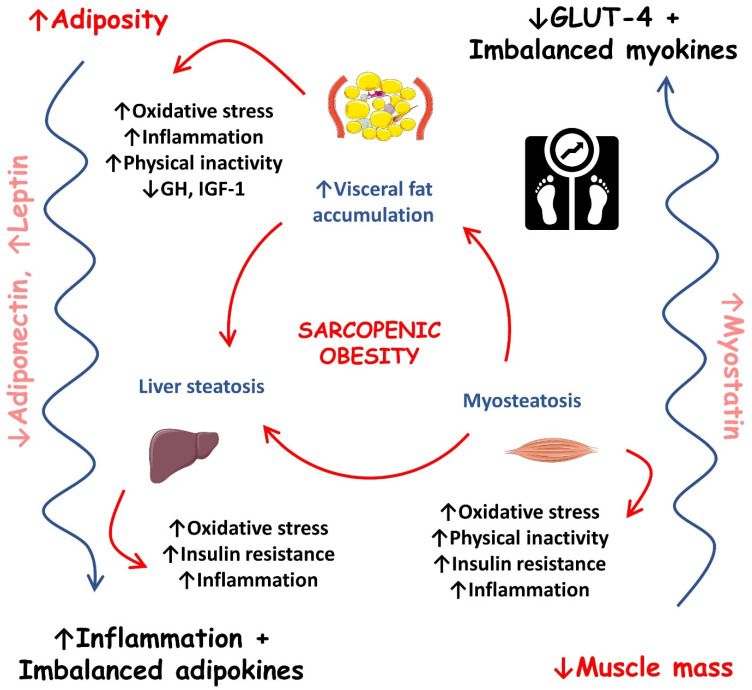
Most important events during sarcopenic obesity development. ↑, increase; ↓, decrease; GH, growth hormone; GLUT-4, solute carrier family 2/facilitated glucose transporter member 4; IGF-1, insulin-like growth factor 1.

**Figure 3 ijms-23-13452-f003:**
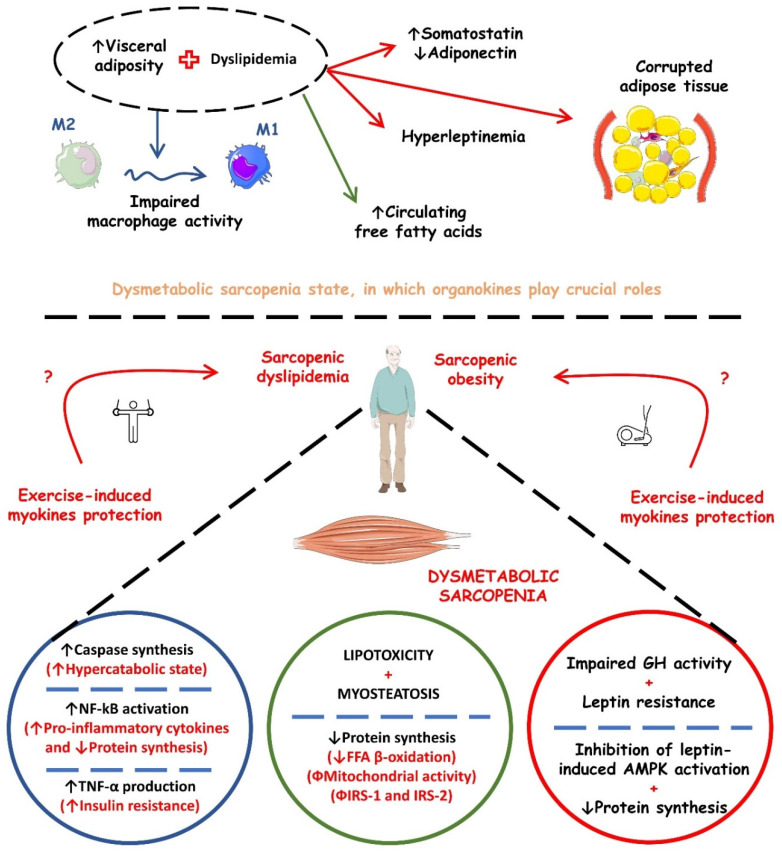
Possible mechanistic associations between organokines secretion and the occurrence of dysmetabolic sarcopenia. ↑, increase; ↓, decrease; AMPK, AMP (adenosine monophosphate)-activated protein kinase; FFA, free fatty acids; GH, growth hormone; IRS-1, insulin receptor substrate 1; IRS-2, insulin receptor substrate 2; NF-kB, nuclear factor kappa b; TNF-α, tumor factor necrosis alpha.

**Figure 4 ijms-23-13452-f004:**
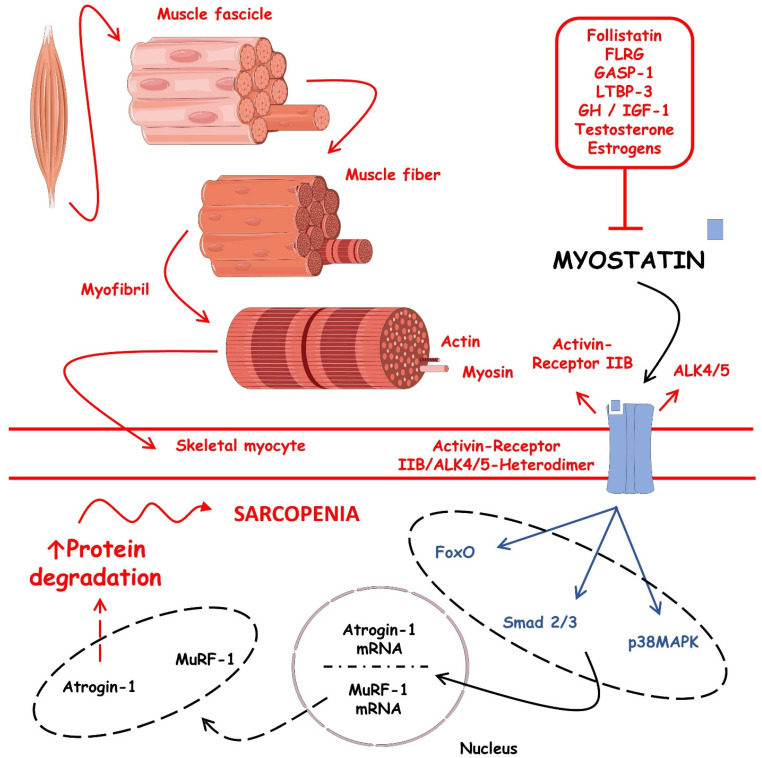
Intracellular signaling pathways involved in the myostatin regulation of protein degradation during sarcopenia. ↑, increase; ALK4/5, anaplastic lymphoma kinase 4/5; FoxO, forkhead box O proteins; FLRG, follistatin-related gene-derived molecule; GASP-1, G-protein coupled receptor-associated sorting protein; GH, growth hormone; IGF-1, insulin-like growth factor 1; IIB, HAD-superfamily hydrolase; mRNA, messenger RNA; MuRF-1, muscle ring-finger protein-1; LTBP-3, latent-transforming growth factor beta-binding protein 3.

**Table 1 ijms-23-13452-t001:** Main characteristics of the organokines involved in the pathophysiology of sarcopenia, diabetes, sarcopenic obesity, and dyslipidemia.

Classification	Organokine	Role in Sarcopenia and Metabolic Repercussions	Expression in Sarcopenia	General Actions	Roles in Sarcopenia	References
MYOKINES	Irisin	Dyslipidemia, sarcopenic obesity, and diabetes.	↓	In ↑ concentrations: ↓ body weight and↑ insulin sensitivity.	In ↓ concentrations, it predicts sarcopenic obesity and ↑ insulin resistance. It is inversely correlated with the occurrence of dyslipidemia.	[68,69]
	Myostatin	Sarcopenic obesity and diabetes.	↑, it is inversely proportional to skeletal muscle mass.	Generates muscle waste and correlates positively with a pro-inflammatory.	In ↑ concentrations, sarcopenic obesity and its driven diabetes can be predicted by myostatin levels. Additionally, it impairs muscle growth and contributes to muscle insulin resistance.	[70,71,72]
	Sclerostin	Sarcopenia and diabetes.	↓	Formation and regulation of bone mass and decrease in bone mass in diabetic patients.	In ↑ concentrations, it is significantly associated with a lower risk of sarcopenia. Hyperglycemia directly increases sclerostin production.	[73,74,75,76]
	BDNF	Dyslipidemia, sarcopenia, obesity, and diabetes.	↓	↑ Memory, ↑ insulin sensitivity, regulates lipid metabolism, and acts in myogenesis process.	In ↓ concentrations, it predicts sarcopenia, obesity, ↑ insulin resistance, and promotes dyslipidemia.	[77,78,79,80]
	IL-6	Sarcopenia, sarcopenic obesity, metabolic syndrome, and diabetes.	↑	↑ Insulin resistance, ↑ inflammation, ↑ muscle volume, and strength.	In ↑ concentrations, sarcopenic obesity, metabolic syndrome, and diabetes, as long as IL-6 is a pro-inflammatory factor.	[10,25,81,82]
	IL-15	Sarcopenia, sarcopenic obesity, obesity, and oxidative stress.	↓	↓ Inflammation, ↓ oxidative stress, ↓ obesity, and ↓ glucose levels.	Prevents the reduction of muscle mass, increases glucose uptake by the skeletal muscle, promotes muscle hypertrophy, and reduces subcutaneous fat and adipose tissue mass.	[10,80,83,84,85]
ADIPOKINES	Leptin	Sarcopenic obesity.	↑	↑ Lipolysis and ↑ insulin sensitivity by skeletal muscles.	Risk predictor for sarcopenic obesity. In ↓ concentrations, leptin can induce insulin resistance by decreasing glucose consumption by muscles.	[86,87].
	LCN2	Sarcopenic obesity and diabetes.	↑	↑ Insulin resistance, ↑ inflammation, and ↑ oxidative stress.	In ↑ concentrations, sarcopenic obesity involves metabolic disorders such as obesity, insulin resistance, and type 2 diabetes.	[25,88,89,90]
	IL-6	Sarcopenia, sarcopenic obesity, metabolic syndrome, and diabetes.	↑	↑ Inflammation and ↑ insulin resistance by inhibiting the expression of IRS1 and GLUT4 in adipocytes.	In ↑ concentrations, sarcopenic obesity, metabolic syndrome, and diabetes.	[10,79,81,91]
	IL-10	Sarcopenia, sarcopenic obesity, obesity, oxidative stress, and autoimmune diabetes.	↓	↓ Inflammation, oxidative stress, and obesity in skeletal muscle.	In spite of an increase in the levels of IL-10 in older humans, in ↑ concentrations it promotes an anti-inflammatory environment and induces protein synthesis in individuals with sarcopenic obesity.	[10,72,92]
	IL-15	Sarcopenia, sarcopenic obesity, obesity, and oxidative stress.	↓	↓ Inflammation, ↓ oxidative stress, ↓ obesity, and ↓ glucose levels.	Prevents the reduction of muscle mass, promotes muscle hypertrophy, and reduces glucose levels, subcutaneous fat, and adipose tissue mass.	[10,80,85,93]
	Apelin	Sarcopenia, obesity, and diabetes.	↓	↓ Inflammation, ↓ glucose levels, and ↓ insulin resistance index.	Improves muscle function in an anti-inflammatory way, increases the regenerative abilities of muscles, and also improves metabolism in obesity and DM2.	[10,80,94]
	IGF-1	Sarcopenia, sarcopenic obesity, and oxidative stress.	↓	↑ Muscle hypertrophy and ↓ atrophy.	In ↓ concentrations, it promotes sarcopenia through ↓ protein synthesis, autophagy, and ↓ muscle regeneration. It also promotes sarcopenic obesity.	[95,96,97] (Naranjo et al., 2017)
	FGF-21	Sarcopenia, sarcopenic obesity, obesity, and oxidative stress.	↓	Regulates glucose and lipid metabolism, antioxidant, ↓ obesity, and ↓ sarcopenia.	In ↓ concentrations, it promotes sarcopenia and obesity since it is directly linked with muscle mass along with irisin.	[10,80,83,98]
	Adiponectin	Sarcopenia, sarcopenic obesity, obesity, and oxidative stress.	↑	↓ Inflammation, ↓ atherosclerosis, ↓ oxidative stress, and ↑ insulin resistance.	In ↑ concentrations, it promotes muscle hypertrophy and ↓ sarcopenic obesity. At the muscle level, ↑free fatty acid oxidation and glucose uptake. At the liver level, ↓ gluconeogenesis.	[10,99]
HEPATOKINES	Fetuin A	Sarcopenic obesity and diabetes.	↑	↑ Insulin resistance, ↑ inflammation, and ↑ oxidative stress.	In ↑ concentrations, it contributes to a state of obesity, sarcopenia, sarcopenic obesity, and insulin resistance through a pro-inflammatory scenario.	[25,100,101,102]
	SHBG	Sarcopenia, sarcopenic obesity, diabetes, and CVD.	↓	↑ Inflammation, ↑ oxidative stress, ↑ obesity, ↑ fatty liver disease, and ↑ CVD.	↑ Levels of SHBG are related to sarcopenia in both sexes of older individuals.	[25,103,104]
	LECT-2	Insulin resistance and obesity.	Probably ↑	In ↑ concentrations: ↑ insulin resistance, obesity, and NAFLD.	Although LECT-2 has no known implications for the mechanism of sarcopenia, it acts directly on skeletal muscle by positively correlating with insulin resistance and obesity.	[99,105,106]
	IGF-1	Sarcopenia, sarcopenic obesity, and oxidative stress.	↓	↑ Muscle hypertrophy, ↓ atrophy, and acts on oxidative stress.	In ↓ concentrations, ↓ protein synthesis, autophagy, and ↓ muscle regeneration, and it promotes sarcopenia and also sarcopenic obesity.	[10,95,107]
OSTEOKINES	Osteocalcin	Sarcopenia, sarcopenic obesity, and diabetes.	↓	In ↑ concentrations: ↑ insulin sensitivity, uptake of glucose, anti-inflammatory, and ↑ muscle mass.	In ↑ concentrations, it contributes to the survival and function of pancreatic β cells, thereby increasing insulin secretion. It also contributes to muscle hypertrophy and strength. Consequently, its reduction would promote the opposite effect.	[108,109]
	Irisin	Dyslipidemia, sarcopenic obesity, and diabetes.	↓	In ↑ concentrations: ↓ body weight and↑insulin sensitivity.	In ↓ concentrations, it predicts sarcopenic obesity and ↑ insulin resistance.	[33,108,110,111].
	Sclerostin	Sarcopenia and diabetes.	↓	Formation and regulation of bone mass and decreased bone mass in diabetic patients.	In ↑ concentrations, it is prospectively associated with a lower risk of sarcopenia.	[73,75,76,112]

↑, increase; ↓, decrease; BDNF, brain-derived neurotrophic factor; IL, interleukin; LCN-2, lipocalin 2; IFG-1, insulin-like growth factor hormone; FGF-21, fibroblast growth factor 21; SHBG, sex hormone binding globulin; CVD, cardiovascular disease; LECT-2, leukocyte-derived chemotaxin-2; NAFLD, non-alcoholic fatty liver disease.

## Data Availability

Not applicable.

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
