# Peer review of "Organokines, Sarcopenia, and Metabolic Repercussions: The Vicious Cycle and the Interplay with Exercise"

_ijms, 2022, doi:10.3390/ijms232113452_

Round 1
Reviewer 1 Report
International Journal of Molecular Sciences
Manuscript ID: ijms-2000772
Type of manuscript: Review
Title: Organokines, Sarcopenia and Metabolic Repercussions: The Vicious Cycle and the Interplay with Exercise
Dear Authors,
Congratulations on writing such an interesting review of Sarcopenia and Metabolic factors. This review demonstrated the comprehensive description of the relationship between muscle, nerve, adipose tissue, and other factors related to sarcopenia, and describes the various hormones as factors linking obesity and sarcopenia. The contents of this review are interesting and well summarized for understanding the relationship between lifestyle-related diseases and the development of sarcopenia. However, there are several problems that listed below.
The following are my comments and suggestions:
1) This paper is “Review”, but your submission style is the form of “regular article”, which seems an inappropriate style. There is no need for a "Results" or "Materials and methods" section in the review. Also, the lack of references cited in “Results” section misleads the results are getting from the authors' experiments. The entire paper needs to be reorganized.
2) Fig. 1 should be a little less wordy. Also, Table.1 should be more compact.
3) It is difficult to understand which references indicate which contents in the main text. Please put the numbers in the appropriate position in the text, instead of summarizing the numbers at the end of the paragraph.
4) Some of the studies in this review seem to be a mixture of human subjects and animal studies or cultured cell lines. I think it is appropriate to describe the studies on human subjects separately from the other studies. At the very least, it should be clearly stated what the studies are on, human or animals.
5) I find the order of the factors listed in section 3.5 confusing. It is unclear which interleukins are divided into myokines and adipokines, and which factors are listed in multiple tissues, such as muscle and adipose tissue.
6) It is also somewhat confusing that irisin and sclerostin have two entries. We think that the number of items needs to be organized, such as describing the contribution to sarcopenia and intervening factors for each tissue such as muscle, adipose tissue, and bone tissue, rather than myokines and adipokines.
Author Response
Dear Doctor,
Please, find below the point-by-point responses to your comments. The modifications in the manuscript are highlighted in yellow.
Comment from Reviewer 1:
Dear Authors,
Congratulations on writing such an interesting review of Sarcopenia and Metabolic factors. This review demonstrated the comprehensive description of the relationship between muscle, nerve, adipose tissue, and other factors related to sarcopenia, and describes the various hormones as factors linking obesity and sarcopenia. The contents of this review are interesting and well summarized for understanding the relationship between lifestyle-related diseases and the development of sarcopenia. However, there are several problems that listed below.
Response: Dear Doctor, thank you very much for your time reviewing our manuscript. We know your time is precious.
The following are my comments and suggestions:
- This paper is "Review", but your submission style is the form of "regular article", which seems an inappropriate style. There is no need for a "Results" or "Materials and methods" section in the review. Also, the lack of references cited in "Results" section misleads the results are getting from the authors' experiments. The entire paper needs to be reorganized.
Response: Dear Doctor, thank you very much for your valuable comment. We reorganized the manuscript. Please, see that we removed the Results sections and reorganized the topics. Please see in the topics that begin in lines 122, 133 and 693.
- 1 should be a little less wordy. Also, Table.1 should be more compact.
Response: Dear Doctor, thank you very much for your observation. We removed the excess words in Figure 1. Please, see lines 210-215.
- It is difficult to understand which references indicate which contents in the main text. Please put the numbers in the appropriate position in the text, instead of summarizing the numbers at the end of the paragraph.
Response: Dear Doctor, thank you very much for your comment. We preferred to leave the references at the end of each sentence since the cited authors have described information related to the whole context.
- Some of the studies in this review seem to be a mixture of human subjects and animal studies or cultured cell lines. I think it is appropriate to describe the studies on human subjects separately from the other studies. At the very least, it should be clearly stated what the studies are on, human or animals.
Response: Dear Doctor, thank you very much for your valuable suggestion. We have included in the text when the studies were with humans, in vivo and in vitro (Please, see lines 155-157; 248-250; 261-262, 275-281; 308-311; 317-319; 323-327; 419-422; 438-440; 452-455; 475-477; 524-528; 532-533; 554-557; 568-575; 602-603; 602-609; 612-616; 647-652, and 658-662). In some cases, there are descriptions of studies in the three categories when we are talking about organokines individually because the intention was to show what exists in the literature about the effects these molecules can exert in sarcopenia and in the metabolic- related diseases.
- I find the order of the factors listed in section 3.5 confusing. It is unclear which interleukins are divided into myokines and adipokines, and which factors are listed in multiple tissues, such as muscle and adipose tissue.
Response: Dear Doctor, thank you very much for your comment. Please, see in line 348 that we first describe Myokines (subtopics 3.5.1.1 to 3.5.1.6); in line 468 Adipokines (subtopics 3.5.2.1 to 3.5.2.8), Hepatokines in line 690 (subtopics 3.5.4.1 to 3.5.4.3) and osteokines (subtopics 3.5.4.1 to 3.5.4.3). However, there are mediators that belong to more than one class, for example, they can have both myokines and adipokines. As an example, we can cite irisin, first described in the literature as a myokine are after as an osteokine.
6) It is also somewhat confusing that irisin and sclerostin have two entries. We think that the number of items needs to be organized, such as describing the contribution to sarcopenia and intervening factors for each tissue such as muscle, adipose tissue, and bone tissue, rather than myokines and adipokines.
Response: Dear Doctor, thank you very much for your comment. We thought of describing each hormone separately as myokine, hepatokine, osteokine, and adipokine so that readers who needed a review of all the molecules would be able to use only one article. As for the description of the same molecule repeatedly in more than one class of organokines, it is due to the fact that they can be released by different tissues. The role of each organokine and the role in sarcopenia and each tissue, such as muscle, adipose tissue, and bone tissue, was performed in Table 1.
Dear Doctor, thank you again for your valuable contributions to improving this manuscript. We are sure that it will be much better after your corrections.
With best regards and wishes for a very nice day!
Reviewer 2 Report
Study review : Organokines, Sarcopenia and Metabolic Repercussions: The Vicious 3 Cycle and the Interplay with Exercise
Very interesting work, but requiring some corrections.
1. in the introduction, there is no information about the role of the selected organokines, myokines, adipokines, osteokines, and hepatokines in the body and the justification of why these were taken into account in the analysis?
Materials and Methods 1. maybe it is worth considering including information from what period these publications came from?
The discussion should be rebuilt in such a way that it constitutes a reliable response to the goals and explains the discussed successes 154-164 these descriptions should be included in the introduction because the results did not mention age. 165-172 as above
Author Response
Dear Doctor,
Please, find below the point-by-point responses to your comments. The modifications in the manuscript are highlighted in yellow.
Comments from Reviewer 2:
Very interesting work, but requiring some corrections.
Response: Dear Doctor, thank you very much for your time reviewing our manuscript. We know your time is precious.
- in the introduction, there is no information about the role of the selected organokines, myokines, adipokines, osteokines, and hepatokines in the body and the justification of why these were taken into account in the analysis?
Response: Dear Doctor, thank you very much for your valuable comment. Please, see in the Introduction section (lines 77-82 and lines 109-115) the justification of why the organokines were investigated in this review.
Materials and Methods 1. maybe it is worth considering including information from what period these publications came from?
Response: Dear Doctor, thank you very much for your important suggestion. We included a statement in lines 130-131 about the period included in the search.
The discussion should be rebuilt in such a way that it constitutes a reliable response to the goals and explains the discussed successes 154-164 these descriptions should be included in the introduction because the results did not mention age. 165-172 as above
Response: Dear Doctor, thank you very much for your suggestion. We included the content of lines 154-164 and 165-172 to the Introduction section, as you may see in lines 107-121. We would like to preserve the sequence used in the Discussion, as it contains a description of each organokine according to the tissue of release. We start with myokines, and then describe adipokines, hepatokines and osteokines. The articles present in the literature are not very clear about the function and activity of these molecules described in a clear way. Therefore, our objective was to describe these organokines and, after that, to include in the table their relationship with sarcopenia and other metabolic comorbidities. We kindly ask that you agree to leave it as it is.
Dear Doctor, thank you again for your valuable contributions to improving this manuscript. We are sure that it will be much better after your corrections.
With best regards and wishes for a very nice day!
Round 2
Reviewer 1 Report
International Journal of Molecular Sciences
Manuscript ID: ijms-2000772
Type of manuscript: Review
Title: Organokines, Sarcopenia and Metabolic Repercussions: The Vicious Cycle and the Interplay with Exercise
After a careful review of this resubmitted manuscript, all my and other reviewer`s comments of the weakness in previous version are clearly improved in this version. So, I recommended this review to be published in the International Journal of Molecular Sciences.
Minor point:
1) The size of the figures should be the same size as Fig.1